# Endotoxemia Associated with Liver Disease Correlates with Systemic Inflammation and T Cell Exhaustion in Hepatitis C Virus Infection

**DOI:** 10.3390/cells12162034

**Published:** 2023-08-10

**Authors:** Carey L. Shive, Corinne M. Kowal, Alexandra F. Desotelle, Ynez Nguyen, Sarah Carbone, Lenche Kostadinova, Perica Davitkov, Megan O’Mara, Alexandra Reihs, Hinnah Siddiqui, Brigid M. Wilson, Donald D. Anthony

**Affiliations:** 1Cleveland VA Medical Center, Cleveland, OH 44106, USA; corinne.kowal@va.gov (C.M.K.); alexandra.desotelle@va.gov (A.F.D.); ynez.nguyen@va.gov (Y.N.); sarah.carbone@va.gov (S.C.); lenche.kostadinova@va.gov (L.K.); perica.davitkov@va.gov (P.D.); megan.omara@va.gov (M.O.); ahr51@case.edu (A.R.); hinnah.siddiqui@va.gov (H.S.); brigid.wilson@va.gov (B.M.W.); dda3@case.edu (D.D.A.); 2Pathology Department, School of Medicine, Case Western Reserve University, Cleveland, OH 44106, USA; 3Department of Medicine, School of Medicine, Case Western Reserve University, Cleveland, OH 44106, USA

**Keywords:** 3-10 HCV, T cell exhaustion, inflammation, EndoCab, LBP, FABP, transient elastography, liver disease

## Abstract

Both acute and chronic hepatitis C virus (HCV) infections are characterized by inflammation. HCV and reduced liver blood filtration contribute to inflammation; however, the mechanisms of systemic immune activation and dysfunction as a result of HCV infection are not clear. We measured circulating inflammatory mediators (IL-6, IP10, sCD163, sCD14), indices of endotoxemia (EndoCab, LBP, FABP), and T cell markers of exhaustion and senescence (PD-1, TIGIT, CD57, KLRG-1) in HCV-infected participants, and followed a small cohort after direct-acting anti-viral therapy. IL-6, IP10, Endocab, LBP, and FABP were elevated in HCV participants, as were T cell co-expression of exhaustion and senescence markers. We found positive associations between IL-6, IP10, EndoCab, LBP, and co-expression of T cell markers of exhaustion and senescence. We also found numerous associations between reduced liver function, as measured by plasma albumin levels, and T cell exhaustion/senescence, inflammation, and endotoxemia. We found positive associations between liver stiffness (TE score) and plasma levels of IL-6, IP10, and LBP. Lastly, plasma IP10 and the proportion of CD8 T cells co-expressing PD-1 and CD57 decreased after initiation of direct-acting anti-viral therapy. Although associations do not prove causality, our results support the model that translocation of microbial products, resulting from decreased liver blood filtration, during HCV infection drives chronic inflammation that results in T cell exhaustion/senescence and contributes to systemic immune dysfunction.

## 1. Introduction

The liver provides both immune surveillance and immune homeostasis [1,2], but when the liver is damaged and cirrhotic or fibrotic, blood pressure in the portal vein increases (portal hypertension), and the filtering function of the liver declines. This results in a decreased ability to filter out gut pathogen-derived molecules such as lipopolysaccharides (LPS). Portal hypertension is also associated with lower platelet counts [3].

Immune responses to pathogen-derived molecules such as LPS can be measured in patient plasma by examining the levels of endotoxin-core antibody (EndoCAb) [4] and lipopolysaccharide-binding protein (LBP), which facilitates the monomerization of LPS. Fatty acid binding protein (FABP) levels may also be an indication of the breakdown of the gut barrier or reduced liver function [5]. Infection with hepatic viruses such as HCV induces chronic inflammation that contributes to liver damage and cirrhosis, with downstream consequences as described above. We and others found that plasma levels of interleukin-6 (IL-6), soluble receptors (sCD14, sCD163), and interferon gamma-induced protein 10 (IP10) [3,6,7,8] were elevated in people with HCV infection. 

Much like chronic inflammation, immune senescence is also characteristic of both chronic viral infections and aging [9,10,11,12]. T cell expression of inhibitory markers programmed cell death protein 1 (PD-1), T cell immunoglobulin and mucin domain-containing protein 3 (Tim-3), and T cell immunoglobulin and ITIM domain (TIGIT) characterize exhausted cells [10,13,14,15], and expression of CD57 and killer cell lectin-like receptor subfamily G (KLRG-1) characterize senescent T cells [16,17]. Cellular exhaustion and senescence often result after multiple rounds of proliferation [18]. Therefore, memory T cells that have undergone repeated clonal expansion are often the T cell maturation subsets that express markers of exhaustion and senescence. One potential contributor to this state is the inflammation-inducing microbial products that are a byproduct of impaired liver filtration, and one downstream consequence appears to be poor vaccine response [8,12] and, more broadly, immune dysfunction. 

If the virus is cleared with direct-acting anti-viral (DAA) therapy, direct virus-mediated inflammation may resolve. However, in the setting of liver cirrhosis and poor liver filtration, inflammation persists. The resulting T cell exhaustion or senescence may be specific to HCV or microbial components, or the T cell exhaustion may be the result of bystander T cell activation [19] and the inflammatory cytokine milieu. Several studies have demonstrated bystander T cell activation during hepatitis infections [20,21,22,23]. How HCV-induced inflammation and liver damage related to filtration impairment impact systemic immune responses in chronic HCV-infected patients is not entirely clear. It is also not clear if endotoxemia or select inflammatory mediators persist after viral clearance.

In this study, we examined plasma from HCV-infected participants, and a small cohort who initiated DAA therapy, for indices of inflammation (IL-6, IP10, sCD163, sCD14) and endotoxemia (Endocab, LBP, FABP). We also measured the expression of T cell markers of exhaustion (PD-1, TIGIT) and senescence (CD57, KLRG-1) and examined the associations of these indices. We measured relationships between liver health (plasma levels of albumin, alanine transaminase (ALT), aspartate transaminase (AST), and transient elastography (TE) score and indices of endotoxemia, inflammation, and T cell exhaustion/senescence. We found that inflammation and endotoxemia were elevated in HCV-infected participants and, with the exception of IP10, did not normalize in our small cohort of patients who initiated DAA therapy. In addition, we found that, in general, T cell exhaustion and senescence were increased in HCV-infected participants and were associated with age. There was a slight decrease in components of T cell exhaustion/senescence after DAA therapy, but our small longitudinal cohort was insufficient to substantiate these results. Lastly, in general, we found that T cell exhaustion/senescence, inflammation, and indices of endotoxemia were positively correlated with each other, as predicted, with some interesting exceptions. While associations cannot demonstrate causality, we believe they can provide information about mechanistic pathways to pursue in future studies. 

## 2. Materials and Methods

### 2.1. Patient Enrollment Ethics Statement 

Informed consent was obtained from all subjects involved in the study. Participant studies were approved by the Cleveland VA Medical Center Institutional Review Board (Approval Code: 1583852) and the University Hospitals of Cleveland Institutional Review Board (Approval Code: 08-98-121).

HCV-infected participants were recruited from the Hepatitis Clinic at the Cleveland VA beginning Spring 2017 through 2018 if they were infected with HCV, had not started DAA therapy, were HIV negative, were over the age of 18, had no current acute viral illness other than HCV, had no immunological disease or immune modulatory treatments within 24 weeks, and were not pregnant or breastfeeding. A small subgroup of HCV patients initiated direct-acting anti-viral therapy (n = 15) and were followed longitudinally. Blood samples were collected once before initiation of DAA therapy and then at 4, 12, 24, and 48 weeks post therapy initiation. They were seen at the Hepatitis Clinic, and follow-up continued through January 2020. 

HCV-uninfected older controls were enrolled through the Cleveland VA geriatric clinics between Spring 2017 and Fall 2020, were >65 years old, HCV and HIV negative, had no current acute viral illness, no immunological disease or immune modulatory treatments within 24 weeks, and were not pregnant or breastfeeding. HCV-uninfected young adult controls were enrolled at the general clinic of the Cleveland VA and at Case Western Reserve University from Spring 2017 to Spring 2019. Uninfected young controls were < or = to 50 years old, HCV and HIV negative, had no current acute viral illness, no immunological disease or immune modulatory treatments within 24 weeks, and were not pregnant or breastfeeding. 

### 2.2. Elisa

Plasma IL-6 was measured by high-sensitivity ELISA (Quantikine HS, R&D Systems, Minneapolis, MN, USA), and soluble CD14 (sCD14), soluble CD163 (sCD163), and Interferon gamma-induced protein 10 (IP10) were measured by ELISA (Quantikine, R&D Systems, Minneapolis, MN, USA). Methods followed the manufacturer’s protocol.

Plasma levels of endotoxin-core IgG antibody (EndoCab IgG) and lipopolysaccharide-binding protein (human LBP) were measured by ELISA (Hycult Biotech, Roelofarendsveen, the Netherlands) following the manufacturer’s instructions. 

FABP2, also known as intestinal fatty acid binding protein (I-FABP), is a member of the cytosolic fatty acid binding protein family. Plasma levels of FABP were measured by ELISA (R&D Systems, Minneapolis, MN, USA) following the general ELISA protocol from the manufacturer.

After the addition of the stop solution, plates were read on a VersaMax absorbance microplate reader and analyzed with SoftMax Pro version 5.4.1 software. 

### 2.3. Cell Preparation

PBMCs were prepared from whole blood by Ficoll–Hypaque density sedimentation and were cryopreserved in 10% dimethyl sulfoxide and 90% FBS until thawing for phenotypic analysis. Longitudinal samples from the same patient were thawed and analyzed in the same experiment.

### 2.4. Flow Cytometry

The T cell phenotype was assessed using the following fluorochrome-conjugated monoclonal antibodies: anti-CD3 PerCP (clone SK7), anti-CD57 FITC (clone NK-1) (BD Biosciences, San Jose, CA, USA), anti-KLRG-1 APC (clone 13F12F2) (eBiosciences, San Diego, CA, USA), anti-CD4 Pacific Blue (clone RPA-T4), anti-CD8 APC-Cy7 (clone SK1), anti-CD45RA PE-Cy7 (clone HI100), anti-CD27 AlexaFluor 700 (clone M-T271), anti-PD-1/CD274 BV711 (clone EH12.2H7), and anti-TIGIT PE (clone A15153G) (Biolegend, San Diego, CA, USA). PBMCs were incubated with viability dye (LIVE/DEAD-Aqua, Invitrogen, Grand Island, NY, USA) at room temperature for 20 min, then washed. Monoclonal antibodies were added for 20 min in the dark at room temperature, washed, and fixed in PBS containing 2% formaldehyde, and events were acquired on a BD LSR Fortessa flow cytometer (Becton Dickinson, San Jose, CA, USA). Data were analyzed using FACSDIVA (version 6.2 BD Bioscience, San Diego, CA, USA) or FlowJo (version 10.5.0) software. T cell maturation subsets were determined based on the expression of CD45RA and CD27; naïve = CD45RA+CD27+; central memory (CM) CD45RA−CD27+; effector memory (EM) CD45RA−CD27−; terminal effector memory (TEM) CD45RA+CD27−. If there were fewer than 100 events in any T cell memory subset, those data were excluded from the analysis. 

### 2.5. Statistics 

Comparisons between two unrelated groups were performed using nonparametric two-tailed Mann–Whitney U tests. Associations between continuous variables were explored by Spearman’s rank order correlation coefficient. Changes after DAA therapy in the HCV longitudinal cohort were measured as changes from pre-therapy (week 0) by the Wilcoxon matched-pairs signed rank test. All statistics were performed using GraphPad version 6, and significance thresholds were set at *p*-values less than 0.05. 

## 3. Results

### 3.1. Patient Characteristics

The median age of the HCV-infected participants (n = 39) was 59 years, ranging from 28 to 74 years old. The uninfected controls were recruited into two age groups: elderly (n = 30), median age 77 years, range 67–89, and young adult (n = 40), median age 34 years, ranging from 23 to 50 years old (Table 1). Most participants were male and uninfected participants were primarily white, while HCV participants were 46% white (Table 1).

Most HCV-infected participants were infected with HCV genotype 1a (19 of 35) with a median viral load of 863,484 IU/mL. The median transient elastography (TE) score was 5.5, and 10 of 35 HCV participants with a documented TE score had a TE score > 12.5, suggesting a high probability of liver cirrhosis, 3 of 35 had a TE score between 9.5 and 12.5, and 22 of 35 had a score < 9.5 (Table 1). Of the 15 HCV patients who initiated DAA therapy, 2 had TE scores >12.5, and 2 did not have TE scores available. All HCV longitudinal patients that started DAA therapy achieved sustained virologic response after 12 weeks of treatment (SVR12).

We were able to collect liver parameters (albumin, alanine transaminase (ALT), aspartate transaminase (AST), and platelet counts) for most of our uninfected elderly (n = 26). Very few of the participants had liver fibrosis or cirrhosis staging scores based on biopsy. We examined alternative non-invasive measurements such as TE scores and AST to platelet ratio index (APRI) calculations to approximate the likelihood of liver fibrosis or cirrhosis [24]. We saw no significant difference in plasma albumin or platelet counts between the HCV-infected participants and the uninfected elderly controls. However, we found elevated plasma levels of ALT (*p* = 0.002) and AST (*p* = 0.0001) in the HCV-infected participants. Although the uninfected elderly cohort was significantly older than the HCV-infected participants, the calculated APRI was significantly elevated in our HCV-infected participants compared to the uninfected elderly participants (*p* = 0.004).

Fifteen of the thirty-nine HCV-infected participants initiated DAA therapy, and we collected blood samples from them before therapy initiation (week 0) and at weeks 4, 12, 24, and 48 post therapy initiation. All demographic and clinical data shown in the patient characteristics table were from the pre-therapy (week 0) time point. 

### 3.2. Indices of Endotoxemia and Systemic Inflammation Are Elevated during HCV Infection

Plasma levels of IL-6 and IP10 were significantly elevated in HCV-infected participants and uninfected elderly compared to uninfected young controls (Figure 1a). Plasma IP10 levels in HCV-infected participants were also elevated when compared to the uninfected elderly (Figure 1a). Plasma sCD163 was elevated in the uninfected elderly compared to the uninfected young but was not significantly elevated in HCV-infected participants (Figure 1a). 

Indices of endotoxemia, LBP, and FABP were significantly elevated in HCV-infected participants compared to uninfected young and elderly controls (Figure 1a). Plasma levels of EndoCab were elevated in HCV-infected participants when compared to the uninfected young participants (Figure 1a), and EndoCab was elevated in the uninfected elderly controls compared to the uninfected young controls (Figure 1a). 

We examined a small cohort of HCV-infected participants longitudinally (pre-DAA therapy, wk4, wk12, wk24, wk48) before and after initiation of direct-acting anti-viral therapy. We saw a significant decrease in plasma levels of IP10 after initiation of DAA therapy but no significant decrease in plasma levels of IL-6, sCD14, or sCD163 (n = 15, Figure 1b). There was no significant decrease in plasma levels of Endocab, LBP, after initiation of DAA therapy in the treated HCV-infected cohort (n = 15, Figure 1b).

### 3.3. Indices of Endotoxemia and Systemic Inflammation Are Associated with T Cell Exhaustion and Senescence Markers during HCV Infection

We examined the co-expression of T cell exhaustion and senescence markers and found that HCV-infected participants had elevated proportions of CD4 and CD8 T cells co-expressing PD-1+TIGIT+ and KLRG-1+TIGIT+ when compared to uninfected young controls (Figure 2a). T cell exhaustion and senescence were even more evident in uninfected elderly. Proportions of CD4 T cells co-expressing PD-1+CD57+, PD-1+TIGIT+, CD57+KLRG-1+, and CD57+TIGIT+ were elevated in the uninfected elderly compared to the young. CD8 T cells co-expressing PD-1+TIGIT+, CD57+KLRG-1+, CD57+TIGIT+, and KLRG-1+TIGIT+ were elevated in the elderly (Figure 2a).

In the cohort of patients that initiated DAA therapy, we saw decreased co-expression of CD57+TIGIT+ on CD4 T cells 4 weeks after initiation of DAA and a decrease in co-expression of PD1+CD57+ on CD8 T cells at 4 and 24 weeks post-DAA therapy initiation (Figure 2b). 

In participants infected with HCV, plasma IL-6 and IP10 levels were positively associated with CD4 T cell co-expression of CD57+TIGIT+, and plasma IP-10 levels were positively associated with CD4 co-expression of PD-1+CD57+. Lastly, plasma levels of sCD163 are positively associated with CD8 T cell co-expression of PD1+KLRG-1+ (Figure 3a). 

In participants infected with HCV, plasma levels of EndoCab were positively associated with CD4 T cell co-expression of PD-1+CD57+ and CD4 T cell co-expression of CD57+KLRG-1+ (Figure 3a). 

Frequencies of T cell markers of exhaustion and senescence vary depending upon the maturation of T cell subsets. Memory T cells, especially terminally differentiated memory T cells, are more likely to express these markers. Therefore, we also examined proportions of each CD4 and CD8 maturation subset (naïve, central memory (CM), effector memory (EM), terminal effector memory (TEM)) that expressed each of the exhaustion (PD-1, TIGIT) and senescence markers (CD57, KLRG-1). We found elevated proportions of CD4 and CD8 T cell expression of PD-1 on naïve T cells from HCV participants compared to uninfected controls, as well as elevated proportions of CD4 T cell expression of TEM PD-1, naïve and TEM TIGIT, and TEM KLRG-1 (Appendix A). Unexpectedly, in HCV participants, we saw lower proportions of CD4 and CD8 T cell expression of CM and EM CD57 and CM KLRG-1; CD4 CM PD-1 and EM TIGIT, and CD8 EM KLRG-1 (Appendix A). Associations of indices in the HCV-infected longitudinal cohort were included with all HCV-infected participants and only examined at the pre-therapy time point.

We would expect that soluble inflammatory markers would contribute to continued T cell activation and, eventually, T cell exhaustion and senescence. Therefore, we would expect to see a positive association between plasma levels of inflammation and T cell markers of exhaustion and senescence. We found that plasma levels of IL-6 were positively associated with CD4 T cell proportions of EM CD57, PD-1, and TIGIT, and TEM CD57, but they were negatively associated with CD4 and CD8 T cell proportions of CM KLRG-1 (Appendix A) during HCV infection. Plasma levels of IP10 positively correlated with CD4 T cell proportions of naïve, EM, and TEM CD57 expression. Soluble CD163 levels were positively associated with proportions of CM CD4 T cells expressing CD57 and naïve CD8 cells expressing PD-1 but negatively associated with proportions of CD4 naïve KLRG-1, CD8 EM, and TEM KLRG-1 expression, and EM TIGIT expression (Appendix A). Also, plasma levels of sCD14 were inversely correlated with CD8 T cell proportions of CM TIGIT (Appendix A).

If our model proposes that poor liver filtration results in increased endotoxemia and inflammation, then we would expect to see a positive association with plasma indices of endotoxemia and T cell exhaustion and senescence. In HCV-infected participants, plasma levels of EndoCab surprisingly correlated negatively with proportions of EM TIGIT+ CD4 T cells but positively correlated with CD8 proportions of EM CD57+ T cells (Appendix A). Plasma levels of LBP negatively correlated with CD4 T cell proportions of naïve and CM KLRG-1 (Appendix A). Lastly, we found that plasma levels of FABP were negatively correlated with CD8 T cell proportions of naïve and CM KLRG-1 and TEM PD-1 cells (Appendix A).

We also found that plasma levels of IL-6 correlated positively with plasma levels of IP10 and LBP during HCV infection. Plasma levels of LBP also positively correlated with plasma levels of EndoCab (Figure 3a).

It is clear that T cell exhaustion and senescence associate with older age. When data from all groups (HCV infected, uninfected >65 yrs, uninfected <45yrs) were combined, age was positively correlated with proportions of CD4 T cells expressing CM CD57 (r = 0.722, *p* = <0.0001), EM CD57 (r = 0.281, *p* = 0.008), naïve TIGIT (r = 0.253, *p* = 0.017), and proportions of CD8 T cells expressing CM TIGIT (r = 0.335, *p* = 0.001), naive TIGIT (r = 0.6525, *p* = <0.0001), naïve CD57 (r = 0.385, *p* = 0.0002), naïve PD-1 (r = 0.487, *p* = <0.0001), and naïve KLRG-1 (r = 0.639, *p* = <0.0001). Age was negatively associated with proportions of CD8 T cells expressing CM KLRG-1 (r = −0.234, *p* = 0.014), EM KLRG-1 (r = −0.310, *p* = 0.004), and EM PD-1 (r = −0.325, *p* = 0.002). 

When we examined the association of age with co-expression of T cell exhaustion and senescence markers, we found that age was positively correlated with proportions of CD4 T cells expressing PD-1+CD57 (r = 0.350, *p* = 0.0008), PD-1+TIGIT (r = 0.317, *p* = 0.003), CD57+KLRG-1 (r = 0.327, *p* = 0.002), CD57+TIGIT (r = 0.339, *p* = 0.001), and proportions of CD8 T cells co-expressing PD-1+CD57 (r = 0.246, *p* = 0.021), PD-1+KLRG-1 (r = 0.258, *p* = 0.015), PD-1+TIGIT (r = 0.476, *p* = <0.0001), CD57+KLRG-1 (r = 0.443, *p* = <0.0001), CD57+TIGIT (r = 0.447, *p* = <0.0001), and KLRG-1+TIGIT (r = 0.599, *p* = <0.0001). 

### 3.4. Plasma Markers of Liver Stress Are Associated with Indices of Endotoxemia, Systemic Inflammation, T Cell Exhaustion, and Senescence during HCV Infection

Next, we examined plasma markers of liver function (albumin) and ongoing inflammation (ALT, AST), as well as liver stiffness as measured by transient elastography (TE). Because albumin is made by liver hepatocytes, albumin levels will be lower if liver function is decreased. Therefore, we would expect that albumin levels would be negatively correlated with inflammation and T cell exhaustion, and senescence. A higher TE score indicates liver stiffness or fibrosis, and we would expect the TE score to positively associate with inflammation and T cell exhaustion/senescence. We found that during HCV infection, plasma levels of IL-6 and IP10 positively correlated with TE score and negatively associated with plasma albumin levels (Figure 3b). Plasma levels of albumin were also negatively associated with age. During HCV infection, plasma levels of EndoCab negatively correlated with plasma levels of albumin, and plasma levels of LBP positively correlated with TE score (Figure 3b).

Plasma levels of albumin during HCV infection were inversely associated with CD4 T cell co-expression of PD1+CD57, PD1+KLRG1, CD57+KLRG1, CD57+TIGIT, and KLRG1+TIGIT (Figure 3b). Plasma albumin levels were also negatively associated with proportions of CD4 CM and EM CD57 (*p* = 0.016, r = −0.417; *p* = 0.009, r = −0.443, respectively) and naïve CD57, PD-1, TIGIT (*p* = 0.002, r = −0.514; *p* = 0.008, r = −0.456; *p* = 0.036, r = −0.366) T cells. Interestingly, plasma albumin levels were positively associated with CD4 CM KLRG1 (*p* = 0.039, r = 0.361), and CD8 proportions of CM and EM KLRG1 (*p* = 0.051, r = 0.342; *p* = 0.002, r = 0.509).

## 4. Discussion

It is clear that HCV infection results in systemic inflammation. A component of this inflammation is caused by the direct immune response to the virus. However, much of this inflammation persists even after clearance of the virus with DAA therapy [7,25]. This persistence may be the result of permanent liver damage and a reduced ability of the liver to filter microbial products from the blood [1,26]. This function is especially important because the liver samples blood that is coming directly from the portal vein that drains the intestine, stomach, spleen, and pancreas [1,2]. Studies that examined microbial translocation in HCV/HIV co-infected patients found that microbial translocation accelerated the progression of liver disease in co-infected patients [27].

We propose that poor liver filtration contributes to elevated levels of inflammation-inducing microbial products and induction of systemic inflammation that further contributes to liver damage, cirrhosis, and continued liver dysfunction. This systemic inflammation may drive a significant amount of bystander T cell activation that results in general T cell exhaustion and senescence [19]. This general T cell exhaustion and senescence may contribute to immune dysfunction and poor vaccine response [8,12].

Our previous studies, and those of others, have demonstrated increased circulating levels of IL-6, IP10, sCD14, and sCD163 in HCV-infected people [3,8,25]. In the current study, we saw significantly elevated plasma levels of IL-6 in the HCV-infected participants. Elevated IL-6 and sCD14 levels are associated with increased mortality in the elderly and in people living with and controlling human immunodeficiency virus (HIV) [28,29,30,31]. CD14 and CD163 are both monocyte markers, and when activated, cells may release soluble forms of the receptors [32,33]. Soluble CD163 is associated with fibrosis in patients with HCV and HBV [32], and sCD163 levels are elevated in patients with acute-on-chronic hepatitis B liver failure and positively associated with end-stage liver disease [34]. In treated patients with HIV, sCD163 levels are associated with non-calcified coronary plaques [35]. As mentioned, sCD14 can be released by activated monocytes [33]. It also may act as an acute reactive protein released by the liver to “soak up” soluble LPS [36,37,38]. In a study examining microbial translocation in women co-infected with HCV/HIV, elevated levels of plasma sCD14, IFABP, and IL-6 were associated with the progression of liver disease [39]. Furthermore, lower levels of sCD14 were associated with early virological response after pegylated-interferon-alpha (peg-INF-alpha) plus ribavirin treatment in HCV/HIV co-infected patients [40].

Previous studies have shown elevated plasma levels of IP10 during HCV infection [6,8,41,42], and pre-PEG IFN therapy, low baseline IP10 is associated with low baseline viral load, a good response to therapy, and less fibrosis, inflammation, and steatosis [42]. Plasma IP10 levels decline after both PEG IFN therapy [42] and DAA therapy [41]. In the current study, we also saw elevated plasma IP10 levels pre-therapy that decreased in a subgroup of patients after initiation of DAA therapy. Interestingly, proportions of co-expression of CD4 T cell TIGIT+CD57+ and CD8 T cell PD-1+CD57+ decreased after DAA therapy in the subgroup of HCV patients, and plasma levels of IP10 positively correlated with CD4 T cell co-expression of TIGIT+CD57+ and PD-1+CD57+. This suggests that IP10 drives the co-expression of CD57, TIGIT, and/or PD-1.

Liver health may be measured in a number of ways. The liver is responsible for the production of acute-phase reactant proteins such as C-reactive protein (CRP) and serum amyloid A (SAA), as well as albumin. Classic clinical measurements of liver inflammation include plasma levels of ALT and AST. Transient elastography (TE) is a non-invasive measure of liver stiffness that relates to liver fibrosis and cirrhosis but also reflects edema. TE uses both ultrasound and elastic waves to generate an image [43]. To measure the filtration function of the liver, studies have measured the translocation of microbial products such as LPS or indications of immune responses to microbial products such as levels of LBP, EndoCab, or FABP [3,4,5,44]. LBP facilitates the binding of LPS to a membrane or soluble CD14, inducing monocyte activation and possibly LPS clearance [38]. FABP is an indication of enterocyte death [3]. In the current study, we found all three parameters (LBP, EndoCab, FABP) elevated in the plasma of the HCV participants compared to the uninfected controls. Importantly, LBP and FABP were elevated in the HCV participants when compared to the uninfected elderly populations as well. This suggests that HCV status and poor liver filtration rather than age drives endotoxemia. One study that examined translocated microbial products and their ability to predict cirrhosis and progression to end-stage liver disease in HCV and HBV infection found elevated plasma levels of LPS, IFABP, IL-6, and sCD14 in patients with HCV or HBV compared to uninfected controls [3]. They also found that sCD14 levels correlated with AST and APRI, and FABP correlated with low platelet counts and inversely correlated with albumin levels [3]. We did not see a correlation between sCD14, AST, or APRI, nor between FABP and albumin levels in the current study. The study mentioned above examined 16 patients with minimal fibrosis and 68 patients with cirrhosis, and sCD14 levels were significantly higher in those with severe fibrosis/cirrhosis. Based on TE scores >12.5, only 10 of the 35 HCV participants with TE scores in the current study likely had cirrhosis. This may explain the lack of similar correlations in the current study compared to the study discussed above.

The second half of our proposed model suggests that systemic inflammation also drives T cell exhaustion and senescence resulting in poor vaccine response and immune dysfunction. The current study did not look at the response to the vaccine but did look at T cell exhaustion and senescence and its associations with inflammation and endotoxemia. We saw significantly elevated co-expression of CD4 and CD8 T cell PD-1+TIGIT+ and KLRG-1+TIGIT+ in our HCV-infected participants compared to the uninfected young controls. Age alone had a significant impact on the proportions of exhausted and senescent T cells. The uninfected elderly controls were more likely to have higher co-expression of T cell markers of exhaustion and senescence than the HCV-infected participants. However, only 7 of the 39 (18%) HCV-infected participants were over the age of 65 years.

Other studies examining T cell exhaustion and senescence during HCV infection found that PD-1 was up-regulated on HCV-specific cytotoxic T lymphocytes (CTLs), natural killer (NK) cells, and non-HCV specific CD4 and CD8 T cells. Furthermore, patients who had sustained virologic responses after IFN-based anti-viral therapy showed decreases in PD-1 on total CD4 T cells, HCV-specific CTLs, and a subset of NH cells [14]. This group later found that T cell immunoglobulin and mucin domain-containing protein 3 (Tim-3) was elevated on CD4 and CD8 T cells during chronic HCV infection. They could also enhance T cell proliferation and IFNg production in response to HCV antigens by blocking Tim-3 [15]. More recently, a group examining exhausted (TCF1+CD127+PD1+) HCV-specific T cells after DAA therapy found that this subset of T cells was maintained after HCV clearance and, upon stimulation, could be induced to proliferate [45].

We found a surprising number of naïve T cells expressing markers of exhaustion and senescence in both the HCV-infected and uninfected participants (Appendix A). This phenomenon was observed by others, and they suggest that there are higher proportions of CD27+CD45RA+ T cells in the elderly that, upon more stringent examination, are actually not true naïve cells [9,46]. They saw elevated proportions of CD27+CD45RA+ T cells expressing PD-1, and when more stringent criteria were used to characterize naïve T cells (CD127+, CD11a-), the PD-1+ cells were not in the true naïve population [9]. We have evidence suggesting that this elevated population of CD27+CD45RA+CD127+CD11a- T cells may not be limited to elderly populations but also found in patients chronically infected with HCV (current study) or HIV (unpublished data).

For the most part, the associations found in the current study support our hypothesis that (1) endotoxemia positively correlates with inflammation (LBP positively correlated with IL-6) (Figure 3a) and cirrhosis (LBP positively correlated with TE) (Figure 3b) and is negatively associated with liver function (Endocab, IL-6, and IP10 negatively correlated with plasma albumin) (Figure 3b); (2) endotoxemia and inflammation positively correlate with T cell exhaustion and senescence (Endocab positively correlated to CD4+ expression of PD-1+CD57+ and CD57+KLRG-1+) and (IL-6 positively correlated with CD4+ expression of CD57+TIGIT+; IP10 positively correlated with CD4+ CD57+TIGIT+ and CD4+ PD-1+CD57+; sCD163 positively correlated with CD8+ PD-1+KLRG-1+) (Figure 3a); and (3) T cell exhaustion and senescence negatively correlates with liver function (albumin levels negatively correlated with all CD4 T cell co-expression of exhaustion and senescence markers) (Figure 3b).

When we examined the associations between T cell subset markers of exhaustion/senescence and endotoxemia and inflammation, several unexpected negative associations were found (Appendix A). Interestingly, most of the negative associations were when T cells were expressing KLRG-1. Although KLRG-1 expression on T cells was previously shown to reflect senescence [17,47], it is also expressed on natural killer (NK) cells. We have no explanation for this finding currently, but it may be related to other functions of KLRG-1 during chronic HCV infection.

Previously, we and others found associations between inflammatory cytokines and T cell exhaustion and senescence [48,49]. We also demonstrated that in vitro stimulation of PBMCs with inflammatory cytokines IL-6 and IL-1b could drive the expression of PD-1 and CD57 on CD4 T cells after 7 days of stimulation [50].

Although we did not examine vaccine responses in the current study, previously, we demonstrated that elevated plasma levels of IL-6, sCD14, and sCD163 were negatively correlated with HAV/HBV (TwinRix) vaccine antibody responses in HCV-infected patients [8]. Another study examining mechanisms of poor vaccine response to the HAV/HBV vaccine in patients infected with HCV found that CD4 T cell responses to ex vivo stimulation with hepatitis B surface antigen (HBsAg) were lower in the poor vaccine responders and PD-1 expression on CD4 T cells was higher in these HBV vaccine poor responders [12]. Further, blocking the PD-1 pathway in vitro improved T cell activation to HBsAg and anti-CD3/CD28 stimulation in poor vaccine responders [12].

One limitation of the current study was that there was no age-matched uninfected control group. However, we did have both younger (<50 yrs) and older (>65 yrs) uninfected participants, and that helped to identify some of the impact of age on the parameters examined. We also had a group of 15 HCV patients that initiated DAA therapy and successfully cleared the virus. This was powerful enough to make some distinctions about the direct effect virus had on the immune parameters that were examined. Another limitation of the study was that the group followed longitudinally after DAA therapy initiation was limited in size, and we were underpowered to make conclusions about all of the data collected. Although we had samples from all longitudinally followed participants at the pre-therapy time point and at 4 weeks post-initiation, we did not have samples from all participants at all time points. And, while we had enough samples to complete all assays on most participants, sample shortages resulted in missingness for some assays from some participants. Also, we were fortunate to be able to have liver parameters (ALT, AST, PLT, and APRI) on the uninfected elderly participants. However, not all HCV patients had TE scores, and very few had biopsies for liver fibrosis stage measurements. Also, our sample size of participants with liver fibrosis or cirrhosis was small, making it difficult to determine the impact of severe liver dysfunction on immune parameters. Lastly, we were not able to directly measure LPS or liver filtration functionality but did have indices of liver health (ALT, AST, PLT, TE, albumin) on all participants.

Associations cannot prove causality, but they can provide support for hypotheses and pathways to pursue future mechanistic studies. Targeting interventions at multiple strategic points (poor liver function, microbial translocation, inflammation, T cell exhaustion/senescence) may help improve immune function in patients with chronic HCV disease even after the virus has been cleared with DAA therapy.

## 5. Conclusions

Indices of endotoxemia and systemic inflammation are elevated in HCV infection, are associated with each other, and persist after HCV clearance. T cell exhaustion and senescence are elevated during HCV infection and persist after HCV clearance, and this is strongly impacted by age. Plasma levels of IL-6, IP10, and EndoCab positively associate with T cell exhaustion and senescence, and T cell exhaustion and senescence highly correlate with low albumin levels. After DAA therapy, plasma IP10 levels and proportions of CD8 T cells co-expressing PD1+CD57+ decline.

## Figures and Tables

**Figure 1 cells-12-02034-f001:**
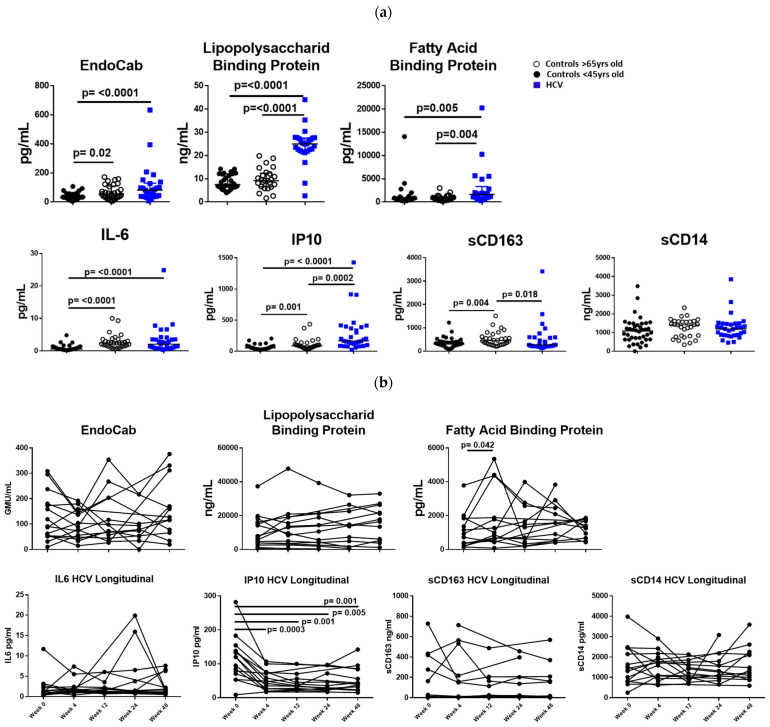
(**a**) **Indices of endotoxemia (EndoCab, LBP, FABP) and systemic inflammation (IL-6, IP10) are elevated during HCV infection**. Plasma levels of endotoxemia (EndoCab, LBP, FABP) and inflammation (IL-6, IP10, sCD163, sCD14) were measured by ELISA. Squares represent HCV-infected participants; closed circles represent uninfected controls <50 years old, and open circles represent uninfected controls >65 years old. Between groups, comparisons were made using Mann–Whitney U tests. *p* values ≤ 0.05 are considered statistically significant. Only statistically significant differences are shown. (**b**) **Plasma IP10 declines after initiation of direct-acting anti-viral therapy**. Plasma levels of endotoxemia (EndoCab, LBP, FABP) and inflammation (IL-6, IP10, sCD163, sCD14) were measured by ELISA pre-DAA therapy and at 4, 12, 24, and 48 after initiation of DAA therapy. Comparison between week 0 and each time point was measured by Wilcoxon matched-pairs signed rank test. Only statistically significant differences are shown.

**Figure 2 cells-12-02034-f002:**
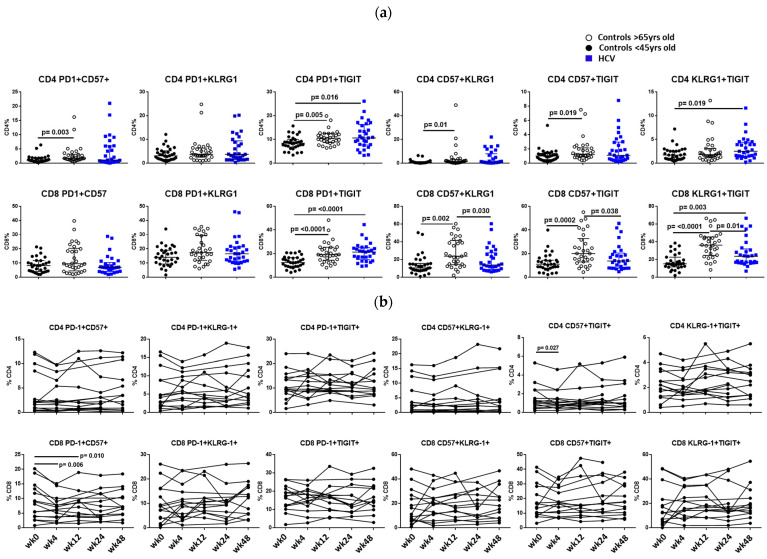
(**a**) **Co-expression of T cell markers of exhaustion and senescence.** Co-expression of surface marker expression of T cell exhaustion (PD1, TIGIT) and senescence (CD57, KLRG1) were measured by flow cytometry. Proportions of live gated, CD3+ CD4+ T cells (top panel) or live, CD3+ CD8+ T cells (bottom panel) co-expressing markers are shown. Squares represent HCV-infected participants; closed circles represent uninfected controls <45 years old, and open circles represent uninfected controls >65 years old. Between groups, comparisons were made using Mann–Whitney U tests. *p* values ≤ 0.05 are considered statistically significant. Only statistically significant differences are shown. (**b**) **Proportion of CD8 T cells co-expressing CD57+PD-1+ declines after initiation of direct-acting anti-viral therapy.** Co-expression of surface marker expression of T cell exhaustion (PD1, TIGIT) and senescence (CD57, KLRG1) were measured by flow cytometry. Proportions of live gated, CD3+ CD4+ T cells (top panel) or live, CD3+ CD8+ T cells (bottom panel) co-expressing markers are shown. Longitudinal samples from the same patient were thawed and analyzed in the same flow cytometry experiment. Samples were tested pre-DAA therapy (week 0) and at 4, 12, 24, and 48 after initiation of DAA therapy. Comparison between week 0 and each time point was measured by Wilcoxon matched-pairs signed rank test. Only statistically significant differences are shown.

**Figure 3 cells-12-02034-f003:**
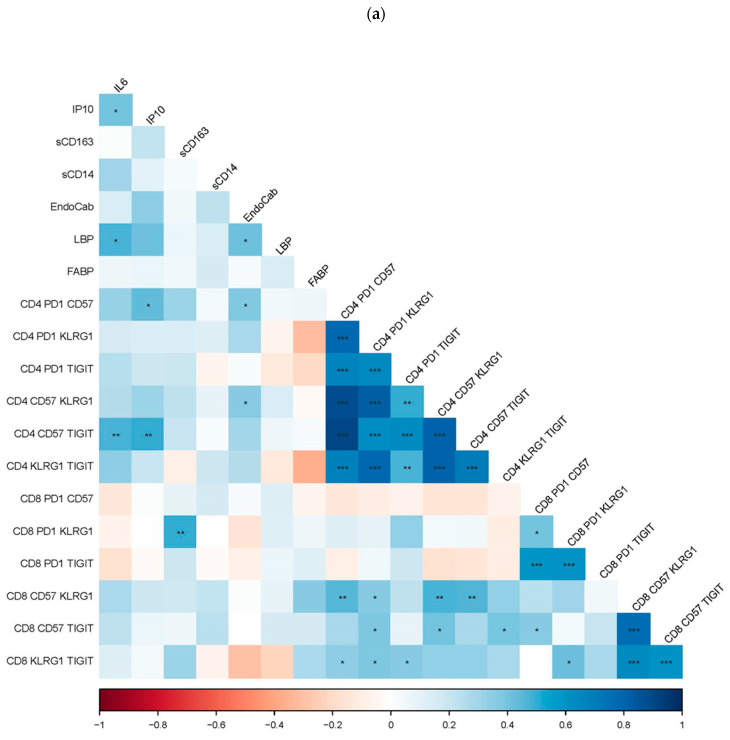
(**a**) **The association of T cell exhaustion and senescence with soluble indices of inflammation and endotoxemia.** The proportions of T cells co-expressing markers of exhaustion and senescence from HCV-infected participants were examined for correlations with levels of soluble markers of inflammation (IL-6, IP10, sCD163, sCD14) and endotoxemia (EndoCab, LBP, FABP) using Spearman’s rank order statistical tests. The r value of correlation is represented by the color and color intensity. Positive correlations are blue, and negative are red. Statistically significant correlations are represented with an asterisk, * *p* = ≤0.05; ** *p* = <0.001; *** *p* = <0.0001. (**b**) **Association of liver health with endotoxemia, inflammation, T cell exhaustion, and senescence during HCV infection.** The proportions of T cells co-expressing markers of exhaustion and senescence and plasma levels of IL-6, IP10, sCD163, sCD14, EndoCab, LBP, and FABP were examined for correlations with indices of liver health, age, albumin, ALT, AST, and transient elastography (TE) score using Spearman’s rank order statistical tests. The r value of correlation is represented by the color and color intensity. Positive correlations are blue, negative are red. Statistically significant correlations are represented with an asterisk, * *p* = ≤0.05; ** *p* = <0.001; *** *p* = <0.0001.

**Table 1 cells-12-02034-t001:** Patient characteristics.

	HCV (n = 39)	Uninfected	
		Elderly (n = 30)	Adult Young (n = 40)
**Age, years median, (min, max), (Q1, Q3)**	59 (28–74) (55, 65)	77 (67–89) (72, 82)	34 (23–50) (28, 43)
**Gender (%)**			
**Male**	94.9	100	71.8
**Female**	5.1	0	28.2
**Race (%)**			
**White, non-Hispanic**	46.2	76.7	94.9
**Black, non-Hispanic**	51.3	23.3	0
**Hispanic (regardless of race)**	2.6	0	2.6
**Indian/Asian**	0	0	2.6
**Albumin median, (Q1, Q3)**	3.7, (3.5, 3.9)	3.7 (3.4, 3.9)	N/A
**ALT median, (Q1, Q3)**	43 (24, 85.5)	26 (21.8, 31.2)	N/A
**AST median, (Q1, Q3)**	29 (20, 51)	18 (16.3, 22.3)	N/A
**PLT median, (Q1, Q3)**	213 (177.3, 262.3)	200 (178, 255)	N/A
**APRI score median, (Q1, Q3)**	0.46 (0.24, 1.50)	0.18 (0.12, 0.24)	N/A
**Transient elastography (TE) (kPa) No/total, (%)**			
**>12.5 kPa**	10/35 (28.6)		
**9.5–12.5**	3/35 (8.6)		
**<9.5**	22/35 (62.9)		
**HCV viral load (IU/mL) median, (Q1, Q3)**	863,484 (42,864, 884,661)	N/A	N/A
**HCV genotype No/total (%)**			
**1**	3/35 (8.6)		
**1a**	19/35 (54.3)		
**1b**	7/35 (20.0)		
**2**	3/35 (8.6)		
**3**	3/35 (8.6)		

min = minimum; max = maximum; Q1 = quartile 1; ALT = alanine transaminase; AST-aspartate transaminase; PLT = platelet count; APRI = AST to platelet ratio index; IU = international units; N/A = not available.

## Data Availability

The data presented in this study are available on request from the corresponding author.

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
