# Peer review of "Endotoxemia Associated with Liver Disease Correlates with Systemic Inflammation and T Cell Exhaustion in Hepatitis C Virus Infection"

_cells, 2023, doi:10.3390/cells12162034_

Round 1
Reviewer 1 Report
The authors investigated in their study ‘Endotoxemia associated with liver disease correlates with systemic inflammation and T cell exhaustion in Hepatitis C virus infection‘ inflammatory mediators in the plasma of HCV infected patients as well as healthy individuals.
The topic of the study is very interesting. However there are several major points which needs to be addressed.
Major points:
1. Comparing HCV infected patients with healthy controls which do not match regarding the age is tricky. The authors needs to comment, why they separated the healthy individuals in 2 groups instead of matching their healthy individuals to their HCV infected patients.
2. In table 1: The authors indicate in their table that they give the range and the IQR. However, it looks like they reported the min-max instead of the range and also the Q1 and Q3 instead of the IQR value. The authors needs to correct this. Also in this table the abbreviations for APRI and N/A is missing.
3. Overall the authors should include all data in their results part, meaning that they show their data at least in the supplementary part as example line 212. Especially the effect of DAA therapy on the investigated parameters is of interest and should play a bigger role in the presentation of the data.
4. The table 2 and 3 are huge and confusing. The authors should consider a different presentation of their data. Also focusing of the important finding in their flow cytometry investigation and moving additional graphs in the supplementary part would be beneficial.
5. The introduction is not connected with the results and the results part is hard to read, since it is just a listing of all their significant findings. The connection of the plasma markers and the analysis of some surface markers on total CD4 and CD8 T cells is not very strong.
6. If the authors would like to strengthen their hypothesis that ‘poor liver filtration contributes to elevated levels of inflammation-inducing microbial products’ the authors should include Patients with liver fibrosis/cirrhosis which are not infected with HCV. Than they would show that their findings are not be induced by the HCV infection.
7. Important citations of the field of HCV are missing.
Minor points:
1. The preparation of the PBMC isolation is missing in the Material and Methods part. Are the staining be performed from fresh or frozen PBMCs? This goes in line with the questions if all samples from one patient were performed on the same day. The authors need to include this in the method part.
2. In the method part the authors wrote ‘Methods followed…protocol.’ For the first 2 ELISAs and gave a briefly description for the 3rd ELISA. The authors should be consistent here.
3. Line 104/105 ‘During…’ The authors need to rephrase the sentence.
4. Line 97: If the University provides an approval number the authors should mention it in this section.
5. APRI needs to be explained.
The english is fine.
Reviewer 2 Report
The manuscript entitled “Endotoxemia associated with liver disease correlates with systemic inflammation and T cell exhaustion in Hepatitis C virus infection” submitted by Shive et al. is a very interesting study into the correlation of inflammation and T cell exhaustion in the context of Hepatitis C infection. The authors do a great job of evaluating a whole host of immunological factors from their patient cohort. A major limitation to the study is the single source patient enrollment and the relatively limited cohort sizes. Despite these limitations there are still very interesting findings that warrant publishing if some of the findings can be clarified.
Considerations to make the manuscript more impactful:
- The group that underwent DAA therapy was said to have significant findings although the data was not shown. This data would be of great interest to see included in the manuscript itself.
- The tables outlining the significant correlative findings are a bit obtrusive and not intuitive. This is due in part to the large open gaps in the tables themselves. Perhaps a list of the parameters evaluated and only show groups that have a significant correlation. This could also be achieved by a heat map with the degrees of color change correlating to the degree of significance.
- It would be of interest to separate the >65 and <45 year old control groups in the dot plots. As was mentioned several times, age plays a role in many of these parameters and it is clear in the figures, these groups are distinctly different. It would be of interest to see how these parameters impact the significance with the HCV positive group.
- Lastly, a summary figure would be of great help to clarify broader trends that were observed in this study.
Reviewer 3 Report
I read with interest the paper entitled “Endotoxemia associated with liver disease correlates with systemic inflammation and T cell exhaustion in Hepatitis C virus infection”. The authors examined the serum concentrations of inflammatory mediators and endotoxemia surrogates in a small cohort of patients with chronic HCV infection. Although the authors address an interesting research question, there are several issues that authors should address to improve the manuscript:
1.) Introduction - The introduction section should be shorter and in parts rewritten.
The purpose of the introduction is to pinpoint the knowledge gap that your study is trying to fill. Avoid presenting textbook knowledge and stick only to what is important for your research. Briefly describe what is already known about the problem and continue logically to what is not known. Do not include your results in Introduction.
2.) Study design is not clear. Include the setting and time-frame of inclusion. Since this is case-control study information on paring should be provided. Why was control group divided in two subgroups depending on age? Please elaborate. In addition, how was the sample size calculated?
3.) Results:
a. The number of participants is unclear – you included 39pts, in 35 had information on the fibrosis stage, and 69 controls? However, you had laboratory data from how many controls?
b. Similarly, in how many patients and controls you measured cytokines.
c. Expand the Table 1 with more data on liver fibrosis stage (F1 to F4)
d. Consider using correlation correlogram instead of tables with correlations coefficients.
e. In Results section avoid discussing your results.
4.) Discussion – You “hypothesize that poor liver filtration contributes to elevated levels of inflammation-inducing microbial products and induction of systemic inflammation that further contributes to liver damage, cirrhosis, and continued liver dysfunction.” However, does your study design and data shown answer to your research question?
5.) Limitations should be added in discussion.
English editing is required.
Round 2
Reviewer 3 Report
The authors have sufficiently responded to my suggestions and significantly improved the manuscript.
I do not have any additional comments.
Minor editing of English language required.